

# Landscape analysis of soil methane flux across complex terrain

Kendra E. Kaiser[1, 2], Brian L. McGlynn[1], and John E. Dore[3, 4]

[1] Earth and Ocean Sciences Department, Nicholas School of the Environment, Duke University, Durham, NC 27708.

[2] Geosciences Department, Boise State University, Boise, ID 83725.

[3] Department of Land Resources and Environmental Sciences, Montana State University, Bozeman, MT 59717.

[4] Montana Institute on Ecosystems, Montana State University, Bozeman, MT 59717.

*Correspondence to*: Kendra E. Kaiser (kendrakaiser@boisestate.edu)

**Abstract.** Relationships between methane ($CH_4$) fluxes and environmental conditions have been extensively explored in saturated soils, while in aerated soils, the relatively small magnitudes of $CH_4$ fluxes have made research less prevalent. Our study builds on previous carbon cycle research at Tenderfoot Creek Experimental Forest, Montana to identify how environmental conditions reflected by topographic metrics can be leveraged to estimate watershed scale $CH_4$ fluxes from point scale measurements. Here, we measured soil $CH_4$ concentrations and fluxes across a range of landscape positions (7 riparian,

25 upland), utilizing topographic and seasonal gradients to examine the relationships between environmental variables, hydrologic dynamics, and $CH_4$ emission and uptake. Riparian areas emitted small fluxes of $CH_4$ throughout the study (median: 0.186 µg $CH_4$-C m$^{-2}$ h$^{-1}$) and uplands increased in sink strength with dry down of the watershed (median: -22.9 µg $CH_4$-C m$^{-2}$ h$^{-1}$). Locations with volumetric water content (VWC) below 38% were methane sinks, and uptake increased with decreasing VWC. Above 43% VWC, net $CH_4$ efflux occurred, and at intermediate VWC net fluxes were near zero. Riparian sites had

near neutral cumulative seasonal flux, and cumulative uptake of $CH_4$ in the uplands was significantly related to topographic indices. These relationships were used to model the net seasonal $CH_4$ flux of the upper Stringer Creek watershed (-1.75 kg $CH_4$-C ha$^{-1}$). This spatially distributed estimate was 111% larger than that obtained by simply extrapolating the mean $CH_4$ flux to the entire watershed area. Our results highlight the importance of quantifying the space-time variability of net $CH_4$ fluxes as predicted by the frequency distribution of landscape positions when assessing watershed scale greenhouse gas balances.

## 1 Introduction

Considerable effort has been directed to the study of carbon dioxide ($CO_2$) fluxes in a variety of diverse terrestrial ecosystems using both spatially distributed chamber measurements and eddy covariance methods (e.g. Lavigne et al., 1997; Sotta et al., 2004; Webster et al., 2008; Riveros-Iregui and McGlynn, 2009; Allaire et al., 2012). However, challenges associated with

measuring upland methane ($CH_4$) fluxes (Denmead, 2008; Wolf et al., 2011) have made similar studies less prevalent, despite $CH_4$ being a more potent greenhouse gas (GHG) than $CO_2$. Global $CH_4$ emissions have increased by 47% since 1970 (IPCC, 2014), and though the soil $CH_4$ sink is significantly smaller than its chemical oxidation in the atmosphere, the uncertainty in the size of the soil $CH_4$ sink is on par with the annual atmospheric $CH_4$ growth rate (Kirschke et al., 2013). Despite progress in our understanding of $CH_4$ dynamics in saturated soils, assessing the variability of CH4 fluxes in aerated soils and exploring




how landscape structure influences CH$_4$ fluxes and watershed CH$_4$ budgets has been limited. Topography can create predictable physical redistribution of resources across a landscape, suggesting that these patterns (e.g., soil moisture: Western et al., 1999, temperature: Urban et al., 2000; Emanuel et al., 2010, and soil organic matter and nutrients: Creed and Band, 1995; Mengistu et al., 2014) could produce observable landscape patterns in soil C fluxes (Webster et al., 2008; Riveros-Iregui and McGlynn, 2009; Pacific et al., 2011).

Net soil CH$_4$ flux can be highly variable in space and time, particularly because microbial production and consumption of CH$_4$ can occur simultaneously in the soil profile. Methane is predominantly consumed in aerated upland soils, and produced in saturated or nearly saturated riparian soils. Methanogenesis occurs under anoxic conditions and at low redox potential though two major microbial pathways (CO$_2$ reduction and acetate fermentation) (Hanson and Hanson, 1996; Mer et al., 2001). Under aerobic conditions, methanotrophic bacteria oxidize CH$_4$ to CO$_2$, and anaerobic oxidation of methane (AOM) can also occur in a variety of environments, including forest soils (e.g. Blazewicz et al., 2012; Gauthier et al., 2015). The interactions of local thermodynamics and environmental conditions including soil moisture, temperature, substrate availability, pH, and oxygen status have made it difficult to determine the most influential parameters across ecosystems (e.g. temperate forest, desert: Luo et al., 2013). In addition, the spatial scale of analysis can influence which environmental factors create observed heterogeneity in CH$_4$ fluxes. For example, the microbial dynamics that drive CH$_4$ cycling are influenced by small scale (cms) environmental conditions (e.g. substrate availability and redox state) (Born et al., 1990; Conrad, 1996). However, at larger scales these environmental conditions can be heavily influenced by physical processes such as landscape scale (kms) hydrology (Burt and Pinay, 2005; Lohse et al., 2009), and at still larger scales (100 kms; here we will refer to as "ecosystem scale"), parent material and climate create the setting in which these processes occur (Potter et al., 1996; Tang et al., 2006; Tian et al., 2010).

A spatially explicit understanding of heterogeneity in CH$_4$ fluxes is necessary for appropriate watershed scale budgets (Ullah and Moore, 2011; Bernhardt et al., 2017), particularly in mountainous regions, where the spatial distribution of resources could have a significant influence on the direction and magnitude of CH$_4$ fluxes due to the lateral redistribution of water and substrates caused by convergent and divergent areas of the landscape (Davidson and Swank, 1986; Meixner and Eugster, 1999; Wachinger et al., 2000; von Fischer and Hedin, 2002). Although many studies have quantified the magnitude and variability of CH$_4$ fluxes, they often covered large spatial extents (from transects 10s of meters long to 100s kms) which captured significant environmental gradients at those scales, but sampling locations were generally sparse (Del Grosso et al., 2000; Dalal and Allen, 2008; Yu et al., 2008; Teh et al., 2014; Tian et al., 2014). The smaller scale patterns of CH$_4$ fluxes within these landscapes has not been investigated as thoroughly as ecosystem scale gradients, which could be problematic if those patterns are important for estimating CH$_4$ fluxes (Fiedler and Sommer, 2000; Ullah and Moore, 2011; Nicolini et al., 2013).

Functional landscape elements have proven useful for assessing spatial heterogeneity and influences of scale in hydrology (Wood et al., 1988), ecology (Forman and Godron, 1981), and biogeochemistry (Corre et al., 1996; Reynolds and Wu, 1999).



Functional landscape elements and terrain metrics that represent topographically driven hydrologic gradients have been used to analyze and scale biogeochemical cycles (e.g., carbon: Creed et al., 2002; Riveros-Iregui and McGlynn, 2009; Pacific et al., 2011; nitrogen: Hedin et al., 1998b; Creed and Beall, 2009; Duncan et al., 2013; Anderson et al., 2015; phosphorus: Devito et al., 2000; sulfate: Welsch et al., 2004), but limited analogous work has been done for $CH_4$ consumption. The importance of

5 soil moisture in mediating $CH_4$ fluxes has been shown across ecosystems (Smith et al., 2000; von Fischer and Hedin, 2007), but studies of how this influence is related to, or predictable from, landscape characteristics have been limited (Boeckx et al., 1997; Creed et al., 2013). Continuous topographic metrics such as the topographic wetness index (TWI; a surrogate for water accumulation), could represent hydrologic influences on variables relevant for $CH_4$ fluxes (e.g. redox state, diffusivity of $CH_4$, and $O_2$, and substrate availability). Here, we build on previous research from Tenderfoot Creek Experimental Forest (TCEF)

that has demonstrated how topographic metrics can represent landscape structure and its influence on hydrologic processes (Jencso et al., 2009; Jencso and McGlynn, 2011) and carbon cycling (Riveros-Iregui and McGlynn, 2009; Pacific et al., 2010, 2011). Our objectives were to determine how locally and distally mediated environmental conditions influence $CH_4$ fluxes, and to estimate the net seasonal $CH_4$ balance of the upper Stringer Creek watershed. Spatially distributed measures of soil moisture, groundwater elevation, and landscape position provide the opportunity to investigate spatial patterns of $CH_4$ fluxes,

linking the point scale conditions to watershed scale hydrologic patterns. We suggest these approaches are beneficial for interpolating, scaling, and predicting $CH_4$ dynamics, particularly in complex terrain. We address the following questions to assess spatial and temporal dynamics of $CH_4$ fluxes across this semi-arid, sub-alpine landscape, examine environmental relationships, and estimate net watershed balances:

- How do environmental variables relate to $CH_4$ flux across a subalpine watershed through the growing season?

- How does landscape structure relate to relative magnitude and direction of $CH_4$ fluxes across the landscape?

## 2 Methods

### 2.1 Site description

Tenderfoot Creek Experimental Forest (TCEF; 46.55° N, 110.52 ° W) is located in the Little Belt Mountains of central Montana (Fig.1). This study was conducted in the upper Stringer Creek watershed (394 ha; elevation 2090−2425 m), a sub-watershed

of TCEF. The gentle to steep-gradient slopes (average 15%) and the range of aspect and topographic convergence/divergence in upper Stringer Creek are characteristic of the greater Tenderfoot Creek watershed (Jencso et al., 2009).



**Figure 1**. Map of upper Stringer Creek watershed (394 ha), located in central Montana, showing sampling locations and meteorological towers. Inset shows transects T1 and T2 profiles where site number increases away from the creek on the west and east sides.

The watershed experiences a continental climate with 70% of the 800 mm annual precipitation typically falling as snow from November to May. Growing season length ranges from 45 to 70 days (Schmidt and Friede, 1996), and mean daily summer





temperature is 11 °C (Farnes et al., 1995). Peak snowmelt typically occurs between mid-May and mid-June, and the driest months occur in the late summer and fall (Fig. 2). Summer precipitation rarely causes significant streamflow response (Nippgen et al., 2011).

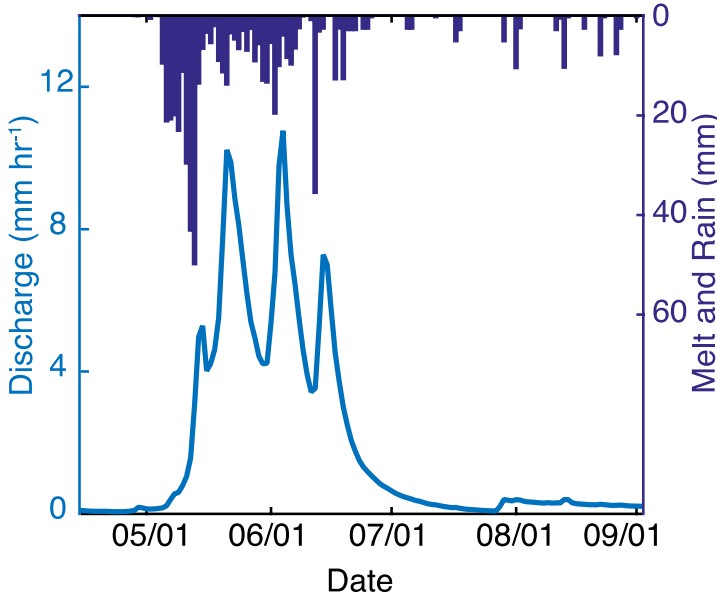

**Figure 2**: Streamflow and precipitation inputs to upper Stringer Creek over the 2013 growing season. Gas sampling began on 29 May, shortly after the first hydrograph peak.

The geology of the Stringer Creek watershed is comprised of Flathead sandstone, Wolsey shale, and granite gneiss. Soils are shallow (<1m) typic cryocrepts in the uplands and aquic cryobalfs in the riparian areas. The seasonal dry down of the upland soils versus the riparian areas (which typically maintain a shallow water table throughout the year; Jencso et al., 2009), reflects the differentiation in soil types. Upland soils have a sandy loam texture, but vary in rock and organic matter content across landscape positions.

Plant communities transition from wet riparian meadows in the valley bottom, through drier meadows to the upland conifer forest. The vegetation in the riparian area is predominately grasses (Juncus, Carex, Poa) and willows (Salix) with a mixture of wildflowers (Erigeron, Aster) The forest is primarily comprised of Lodgepole pine (Pinus contorta), and subalpine fir (Abies lasiocarpa); Englemann spruce (Picea engelmannii) and whitebark pine (Pinus albicaulis) are also common; Grouse whortleberry (Vaccinium scoparium) is dominant in the understory (Mincemoyer and Birdsall, 2006).



## 2.2 Landscape characterization

Ten meter and three meter digital elevation models (DEMs) were constructed by coarsening 1 m$^2$ resolution light detection and ranging (LIDAR) data. These data were collected in 2005 by the National Center for Airborne Laser Mapping (NCALM). We calculated topographic characteristics that describe both energy availability and relative water availability of each site

5   using DEM landscape analysis methods as described in McGlynn and Seibert (2003). Terrain metrics included in the analysis were: aspect (radians), elevation (m), insolation (kWh m$^{-2}$), slope (%), elevation above the creek (EAC, m), distance from creek (DFC, m), gradient to creek (GTC), upslope accumulated area (UAA, m$^2$), and the topographic wetness index (TWI) (Jencso and McGlynn, 2011; Nippgen et al., 2011). Position-on-slope and aspect effects were calculated using the following equations (Clark, 1990):

$$aspect = \begin{pmatrix} \cos\phi\sin\theta \\ \sin\phi\sin\theta \end{pmatrix} \qquad (1)$$

$$slope = \cos\theta \qquad (2)$$

where $\phi$ is aspect and $\theta$ is slope. Potential incoming insolation (from May 1$^{st}$ to September 1$^{st}$) was calculated in the System for Automated Geoscientific Analysis (SAGA) using one hour increments averaged over five day windows (Böhner and Antonic, 2009). UAA is the watershed area contributing to each point in the landscape and was derived using the MD$\infty$

algorithm (Seibert and McGlynn, 2007). TWI has also been used as an approximation for relative wetness and was calculated using the following equation (Beven and Kirkby, 1978):

$$TWI = ln\left(\frac{a}{\tan\theta}\right) \qquad (3)$$

where a is UAA and $\theta$ is local slope. All topographic metrics were assessed for relationships with CH$_4$, and for inclusion in the multivariate model. The riparian area was initially delineated as the area less than two meters in elevation above the creek

using topographically derived flowpaths and validated with extensive field surveys (90 transects; Jencso et al., 2010).

We examined the spatial and temporal variability of CH$_4$ fluxes using data collected across a range of landscape positions in the Stringer Creek watershed. Terrain metrics were used to select 32 sampling sites that span a range of slopes, contributing area, and convergence/ divergence in the upper Stringer Creek watershed (Fig. 1). Twenty-five sites were distributed across the uplands, and two transects that cross Stringer creek (with 3–4 riparian sites each) were selected to characterize the riparian

and lower hillslope positions and their transition. The transition zone between the riparian area and the uplands is identified hydrologically as the toe slope position, where groundwater tables persist longer than in the uplands, but not through the growing season as observed in the riparian area. Measurement sites along the two transects that cross the creek are identified by the side of the creek they are on (East/West) and increase in number away from the creek.



### 2.3 Soil characterization measurements

Soil cores were collected on July 8th and 9th, 2012 within two meters of each gas sampling site for soil analysis. Soil cores were extracted using a 100 cm$^3$ cylinder that was inserted, and laterally excavated from the organic (0−10 cm) and mineral (22.5−27.5 and 47.5−52.5 cm) soil layers (n = 32). The samples were dried and weighed to calculate bulk density and were

analysed for % Carbon (C) and % Nitrogen (N), $\partial^{13}$C, and $\partial^{15}$N (Kansas State Stable Isotope Mass Spectrometry lab, ThermoFinnigan Delta Plus mass spectrometer and CE 1110 elemental analyser with ConFlo II Universal Interface for C and N analysis of solids, additional details in Nippert et al., 2013). These data were used to calculate total $C_{soil}$ (g cm$^{-2}$), $N_{soil}$ (g cm$^{-2}$) and molar C:N ratios. Intact soil cores and bulk soil samples were also collected from 0−5 cm on August 6th and 7th, 2014 to determine porosity, bulk density, mineral bulk density and organic content for each site. Porosity was determined by measuring

the change in weight between fully saturated and oven dried intact soil cores (n = 18). The bulk samples were used to corroborate bulk density and particle size distribution following standard procedures.

### 2.4 Environmental measurements

Weekly measurements of environmental variables were collected in conjunction with gas samples at each site from May− September 2013 between 900 and 1800 hours. Environmental variables that were measured included volumetric soil water

content (VWC), soil temperature (12 cm soil thermometer, Reotemp Instrument Corporation, California), and barometric pressure (Atmospheric Data Center Pro, Brunton, Boulder, CO). Volumetric soil water content (VWC) was measured three times at each site during each round of sampling using a Hydrosense II portable Soil Water Content meter, (12 cm, Campbell Scientific Inc., Utah, United States). The mean of the three samples was used for data analysis. We measured real-time water content hourly at individual riparian (T1E1), transition (T1E2), hillslope (T1E3) sites using water content probes (CSI model

616, Campbell Scientific Inc., Utah, United States) that were inserted from 0−12 cm in the soil (Fig. 1).

### 2.5 Hydrological measurements

Groundwater table data were recorded in wells located along the two riparian-hillslope transects to augment the weekly measurements of near surface soil water content (Fig. 1). Groundwater wells (created from 3.81 cm diameter polyvinyl chloride (PVC), screened from completion depth to within 10 cm of ground surface), were installed along the riparian-hillslope transects

(co-located with gas wells). Capacitance rods (± 1mm, Tru Track, Inc., New Zealand) in each well recorded groundwater level every 30 minutes. Well completion depths (to the soil bedrock interface) ranged from 0.5−1 m in the riparian zones and 0.8− 1.5 m on the hillslopes. Installation details can be found in Jencso et al. (2009). Groundwater data were also used to evaluate our initial delineation of sites as riparian versus upland.



### 2.6 Soil gas measurements and flux calculations

Soil gas wells constructed of 5.25 cm diameter, 15 cm long sections of PVC were installed to sample soil air for concentration measurements of $CH_4$, $CO_2$ and $O_2$ at 5 cm, 20 cm, and 50 cm depths. Gas wells were buried at completion depths of 20 and 50 cm and capped with a size 11 rubber stopper. These gas wells were open at the bottom to equilibrate with soil gas at their

respective depths. Shallow gas wells were designed to measure gas concentrations closer to the soil surface; the bottoms were closed with a PVC cap, and screened openings on the sides enabled equilibration with soil gas at 5 cm depth. All gas wells were outfitted with a closed sampling loop made of PVC tubing (4.8 mm inside diameter, Nalgene 180 clear PVC, Nalgene Nunc International, Rochester New York, USA) that was passed through the rubber stopper, and joined above the ground surface by 6–8 mm HDPE tubing connectors (FisherBrand, Fisher Scientific, USA). Thus, the equilibrium volume was the

volume of the PVC well plus that of the tubing.

Weekly gas samples were taken from the closed recirculation loop after observed soil $CO_2$ concentration stabilized. Soil $CO_2$ concentration was measured in-line using a Vaisala Carbocap handheld $CO_2$ meter (GM70, measurement range of either 0–20,000 ppmv, or 0–50,000 ppmv) adjusted for local temperature and pressure. Soil $O_2$ % was also measured in-line using a

handheld Apogee $O_2$ sensor (MO-200, Logan, Utah; precision ± 0.1 % $O_2$). Once the $CO_2$ concentration reading stabilized, one gas sample was collected from each depth through a brass Swagelok T fitting with a 9.5 mm Cole Palmer Septum (Vernon, IL) sampling port, using a Precision Glide needle (22G1, Becton Dickinson & Co, NJ) and 60 mL Luer-Lok syringe (BD, Franklin Lakes, NJ). Gas samples (~50 mL) were transferred to and stored in 150 mL laminated Flex Foil sample bags (SKC, Eighty Four, PA). Prior to sample collection in the field, sample bags were emptied by vacuum, filled with $N_2$ carrier gas, and

evacuated in the lab to avoid sample contamination. This process was done twice for bags that had previously contained gas with concentrations of $CH_4$ considerably higher than ambient. The sampling syringe was cleared between samples and flushed with 10 mL of air from the gas well three times before slowly taking the sample (to avoid creating any vacuum in the gas well). During snowmelt, saturated soils in the riparian area resulted in flooded wells, preventing gas sampling at those time points.

Gas samples were analysed for $CH_4$ at Montana State University using a Hewlett-Packard 5890 Series II gas chromatograph outfitted with a flame ionization detector (FID). The inlet system used a 10-port injection valve with a 1 cm³ sample loop. The injection valve was configured for backflushing of a precolumn (25 cm x 0.32 cm OD, packed with Porapak-T 80/100 mesh) to prevent water vapor from reaching the analytical column. The sample loop temperature (ambient) was monitored using a NIST-traceable electronic thermometer, and barometric pressure was obtained from the Montana State University weather

station (operated by Dr. Joseph A. Shaw). Two analytical columns (both 183 cm x 0.32 cm OD, packed with Chromosorb 102 80/100 mesh and Porapak-Q 80/100 mesh, respectively) were used in series for gas separation. The temperatures of the column oven and FID were 55 °C and 240 °C, respectively. The carrier gas was a commercial ultra-high purity $N_2$, which was further purified through Molecular Sieve 5A, activated charcoal and an oxygen scrubber. The carrier flow to the FID was maintained





at approximately 30 mL min-1. Under these conditions, $CH_4$ eluted to the FID at 1.9 min. A certified 51 ppmv $CH_4$ in air standard (Air Liquide; ± 1% accuracy) was used for instrument calibration, both alone and after dilution into ultra high purity $N_2$ carrier gas; the detector response was linear and the overall analytical precision was better than ± 0.05 ppmv.

Methane fluxes were calculated using the gradient method (Fick's first law) and measured soil concentrations at 5 cm (Eq. 4).

$$f_{CH4} = CH_4 * \left( \frac{d[CH_4]}{dz} \right) \qquad (4)$$

where $f_{CH4}$ is the flux of $CH_4$ out of the soil (µg $CH_4$-C m$^{-2}$ h$^{-1}$), $D_{CH4}$ is the $CH_4$ effective diffusivity (m$^2$ h$^{-1}$), and $(d[CH_4])/dz$ is the $CH_4$ gradient from 0.05 m to the soil surface (µg $CH_4$-C m$^{-4}$; the distance from the soil surface, z (m), is defined as positive upward). For determination of $(d[CH_4])/dz$, measured mole fractions of $CH_4$ were converted to mass concentrations assuming ideality of gases and using the measured soil temperature. Although this formulation does not include production or

consumption that is occurring between 5 cm and the surface, the $CH_4$ concentration gradient from shallow depths to the surface is typically relatively linear (Koschorreck and Conrad, 1993), suggesting that determining $f_{CH4}$ using a linear equation is appropriate. Effective diffusivity was estimated for each sample using an empirical relationship between the measured VWC and $CH_4$ diffusivity (Fig. 3). This relationship was established by measuring methane flux and concentrations across a variety of spatial locations (co-located with gas wells) and time points using a LI-COR 8100A infrared gas analyser with a 20 cm

diameter chamber and an in-line sampling port for collecting discrete time-course $CH_4$ samples from the chamber. Our exponential model relating effective $CH_4$ diffusivity to soil water content is mathematically equivalent to an exponential fit of diffusivity to air-filled pore space (Richter et al., 1991) when total porosity is treated as a constant. Our model results were in good agreement with other commonly used physical models of soil gas diffusion for total porosities near 0.6 (Buckingham, 1904; Ghanbarian and Hunt, 2014; Møldrup et al., 2014), and incorporate site to site variability due to local VWC. We

calculated cumulative seasonal $CH_4$ flux ($F_{CH4}$) for each site (May 29th–September 12th) by summing the linearly interpolated daily fluxes. We believe that this parsimonious approach is appropriate to assess how landscape position influences the relative magnitude of seasonal $CH_4$ fluxes.





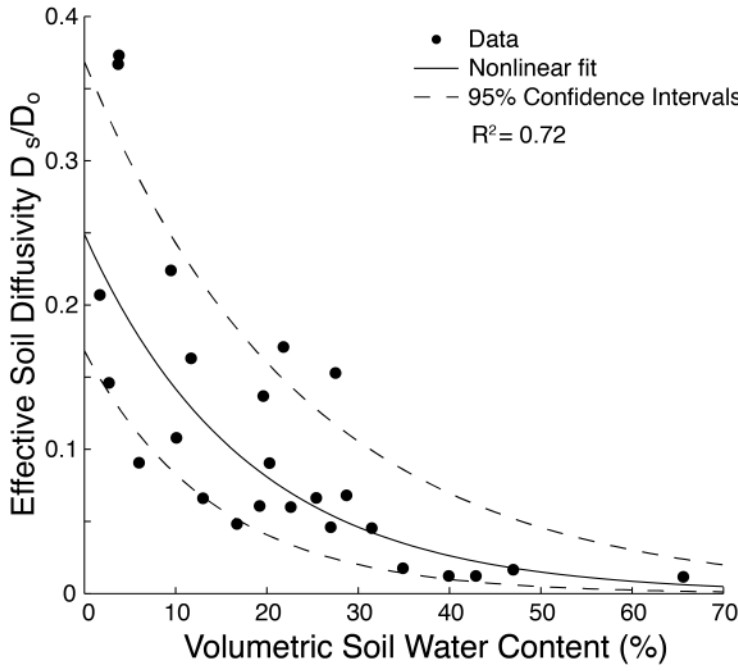

**Figure 3:** Relationship between effective soil diffusivity for methane $D_s$ (expressed relative to its free-air diffusivity $D_o$) and soil water content. This empirical relationship was used to estimate 0-5 cm soil $CH_4$ diffusivity for every sampling event at each site.

### 2.8 Statistics and modelling

We used two-sample t-tests to test for differences between $f_{CH4}$ and environmental variables across riparian and upland locations. We performed linear regression analysis on the upland $f_{CH4}$ fluxes to assess relationships between instantaneous upland $f_{CH4}$ measurements and environmental variables using the R Stats Package (R Core Team, 2016). Further linear

10  regression analysis was performed on natural log-transformed cumulative $CH_4$ influx ($\ln|F_{CH4}|_{in}$) versus all terrain metrics, soil properties, and each site's average VWC ($VWC_{avg}$) and temperature ($T_{avg}$). We log transformed the absolute value of $F_{CH4}|_{in}$ to meet linear regression assumptions of homoscedacity and linearity. If a set of variables had a Pearson's R correlation coefficient > 0.6 (Dorman et al., 2013), the variable with a lower correlation to $\ln|F_{CH4}|_{in}$ was removed from the multiple regression analysis to prevent multicollinearity in the final model (Fig. S1 and A2).

We assessed two sets of predictor variables for multiple regression modelling: 1) both terrain metrics and local soil measurements ($VWC_{avg}$, $T_{avg}$, and soil properties) and 2) only terrain metrics. To remove the potential influence of collinearity, we subset the predictor variables. If a set of topographic or soils variables had a pearson correlation coefficient greater than





0.6, then the variable with a lower correlation with $\ln|F_{CH4}|_{in}$ was removed from the analysis (Fig. A1 and A2). We also assessed the strength of local soil measurements independent of the multiple regression models to determine which local variables were of most importance. A parameter jack-knife method (Phillips, 2006, 2008) was used to determine the importance of individual variables within each set of data (Fig. A4). We used the Leaps package and the exhaustive search method (Lumley and Miller, 2009) to select the best linear multiple regression model using terrain metrics and local measurements, and terrain metrics alone in order to create a spatially distributed estimate of $\ln|F_{CH4}|_{in}$. Model assessment was based on the adjusted $r^2$ and Bayesian Information Criterion (BIC; see Sect. 3.5). Given the necessity of using the same dataset to select predictor variables that were used to create the model, we performed a leave-one-out cross validation (LOOCV) using the DAAG package (Maindonald and Braun, 2015) to determine the mean square error of each model.

## 3 Results

### 3.1 Terrain analysis

Terrain analysis was performed using both 3 m and 10 m DEMs, and although higher resolution mapping can be beneficial in some scenarios, the 10 m flow accumulation results have been shown to be more reflective of the lateral transport of water in TCEF and were used in this analysis (Jencso et al., 2009). The slopes in the upper Stringer Creek watershed range from moderate (2%) to steep (66%). Sampling sites encompassed the range of aspects in the watershed (72°–312°), however the range of potential incoming solar radiation was relatively narrow over the growing season (1026–1141 kWh m$^{-2}$). Our site selection spanned a range of landscape hydrologic settings with UAAs ranging from 318 m$^2$ to 10,667 m$^2$ in the uplands, with one site representing a less frequent but much higher UAA (22,981 m$^2$). This site was removed from upland regression analysis due to the strong leverage it had on observed relationships. Riparian sites were not characterized by the 10 m$^2$ DEM due to their relatively small extent (less than the grid size) and challenges associated with discerning between down hillslope and down valley flow accumulation. A threshold of 2 m in elevation above the creek (EAC) was used to identify riparian areas (Jencso et al., 2009), and was consistent with field observations for 5 of 7 riparian gas well nests. One site (T2W3), located 40 m away from the creek (4.5 m EAC) was heavily influenced by the large upstream riparian extent and gentle local slope, which resulted in it maintaining a groundwater table and high soil water content throughout the season, characteristic of riparian sites. Alternatively, a sampling site that was located within the EAC delineated riparian area (T2W2 1.5 m EAC, 15 m away), no longer had a groundwater table present by late July, and had a steady decline in soil water content which is characteristic of hillslope locations. The hydrologic dynamics of these sites suggested that their CH$_4$ dynamics could be better characterized by categorizing them based on hydrologic measurements rather than the simple terrain analysis.



## 3.2 Range and seasonality of environmental variables

Soil molar C:N ratios ranged from 13−43 in the shallow soil samples (0−5 cm). Average bulk density was 0.64 g cm$^{-3}$ at riparian sites (n = 7) and 0.75 g cm$^{-3}$ in the uplands (n = 25; Table 1). Average soil porosity in the riparian area (0.76; n = 6) was significantly higher (two-sample t-test, p < 0.05) than average soil porosity of the uplands (0.65; n = 12), and agreed well

with the estimated landscape-average soil porosity of ~0.6 implicated by the exponential VWC-diffusivity relationship (Fig. 3).

Soil temperatures ranged from 0° to 8°C across all sampling sites during the first sampling event on May 23$^{rd}$, 2013, and reached the seasonal maximum soil temperature (9−20 °C) by mid-July. Soil temperatures declined through August with

seasonally intermediate temperatures by September 12th (8−15 °C). The average soil temperature in the riparian area (11.5 °C) was higher than that of the upland soils (10.6 °C), likely due to minimal canopy cover and thus higher insolation in the riparian corridor.

Volumetric soil water content (VWC) had a strong seasonal pattern and was significantly different between riparian and upland

landscape positions (two-sample t-test, p < 0.001; Fig. 4), as shown by real time water content probes and spatially distributed VWC measurements (Fig. 5). VWC reached a minimum (2−12% in the uplands and 25−55% in the riparian area) in late July prior to a sequence of late season rain events that increased the range of VWC in the uplands to 3−21%, and the riparian area to 29−59%.

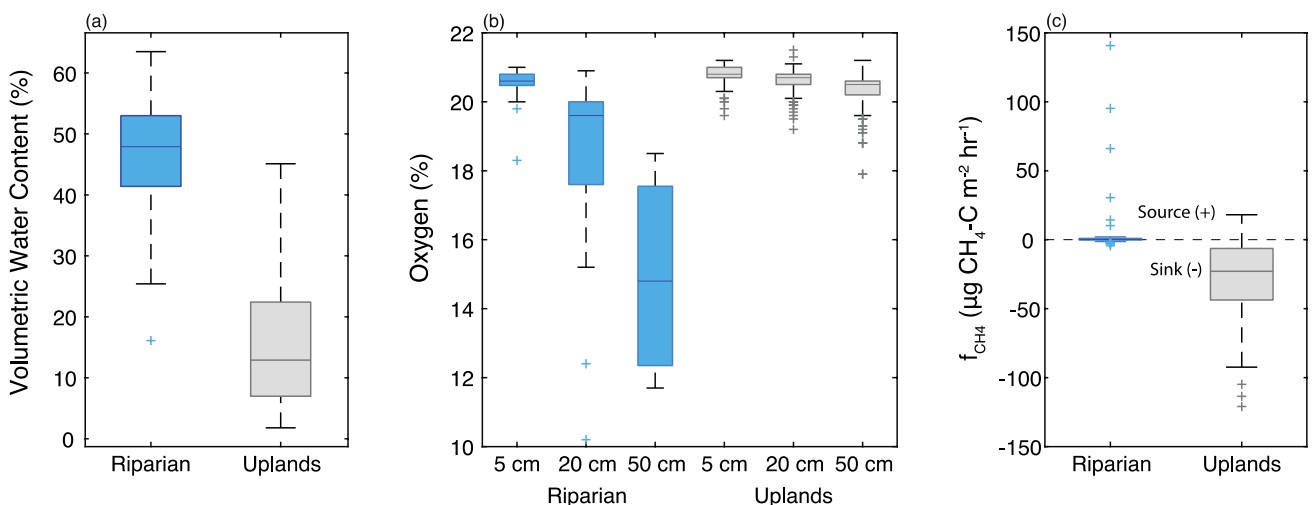

**Figure 4:** (a) Riparian and upland soil water content. (b) Percent oxygen of riparian and upland landscape positions at 5, 20 and 50 cm. (c) Methane flux in riparian and upland landscape positions. Riparian and upland sample sets were significantly different for all sets of data except for the 5cm O₂



data (two-sample t-test $p < 0.01$). Riparian measurements n = 53; upland measurements n = 259. Boxes denote 25[th], 50[th], and 75[th] percentiles, whiskers represent maximum and minimum values, and crosses denote outliers (greater than 75[th] percentile times interquartile range, or less than 25[th] percentile times interquartile range).

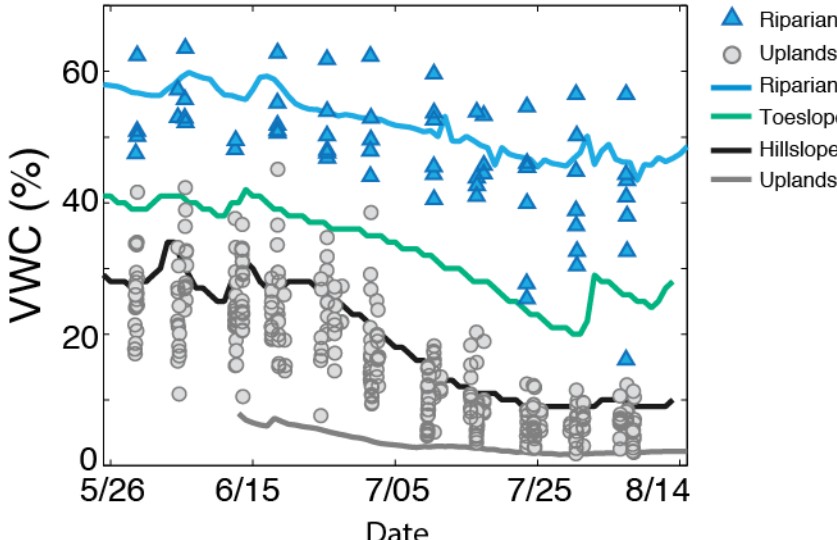

**Figure 5**: Real time water content sensors (solid lines) that were distributed across landscape positions during the growing season of 2013 show the seasonal dry down of the landscape, with a muted signal in the riparian area. These high-frequency sensor data corroborate the distributed volumetric water content (VWC) measurements made at every site during discrete sampling (filled symbols). Riparian sites increase in variability throughout the season, and hillslope positions gradually dry down to low soil moisture conditions.

Groundwater (GW) table dynamics can be described by three general responses that were related to proximity to the creek (Fig. 6). Riparian locations maintained a GW table throughout the season, with near surface saturation during snowmelt, and GW tables 20–50 cm below the soil surface late in the season. GW wells closest to the stream (T1E1 and T1W1) had a water table within 22–25 cm of the surface throughout the season. Toe slope positions (near the strong break in slope on the east side), responded rapidly to snowmelt, and retained a GW table through late July. Wells in this transition zone (e.g., T1E2, Fig. 6c) had variable GW dynamics, which included GW response to the rain events (up to 11 mm) in the first week of August. At another transition location, a well that was influenced by the large local riparian extent, and low local gradient (T2W3) maintained a GW table within 70 cm of the surface throughout the season. Upland positions above the break in slope exhibited transient GW tables during peak snowmelt, and by mid to late June no longer had GW tables present. During snowmelt these wells had a GW table for up to 28 days and no wells had a GW table after 26 June.



The shallow soil was well oxygenated; in the uplands 5 cm $O_2$ ranged from 19.6 to 21.2%, and 5 cm riparian $O_2$ ranged from 18.3 to 21.0% in the soil atmosphere (Fig. 4). Upland soils were well oxygenated across all sites and depths (19.2–21.5% $O_2$ at 20 cm, 17.9–21.2 % $O_2$ at 50 cm; Fig. 4). The only substantial depletion of $O_2$ was in the 20 and 50 cm samples in the riparian area, which ranged from 10.2–20.9% $O_2$ at 20 cm, and 11.7–18.5% $O_2$ at 50 cm (Fig. 4). Median $O_2$ of riparian sites decreased from 20.5% at 5 cm to 18.2% at 20 cm and 16.7% at 50 cm.

Methane fluxes ($f_{CH4}$) exhibited a considerable range across the landscape (-121 to 141 µg $CH_4$-C $m^{-2}$ $h^{-1}$; Fig. 4, Table 2), with significantly different $f_{CH4}$ between the riparian and upland positions (two-sample t-test, $p < 0.001$). Riparian $CH_4$ efflux was generally low, and the upland positions were predominately sinks (Table 2). Upland locations did produce small $CH_4$ fluxes out of the soil (up to 3.5 µg $CH_4$-C $m^{-2}$ $h^{-1}$) early in the season.



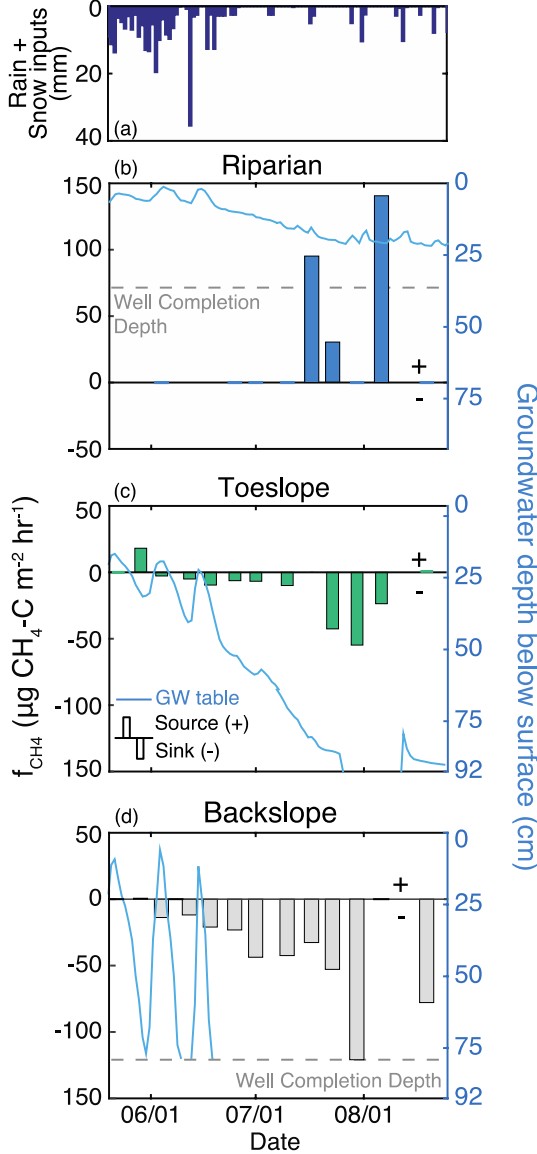

**Figure 6:** Methane dynamics and the seasonal decline of the groundwater (GW) table at three sites located along a riparian-hillslope transect during the 2013 growing season. (a) Rain and snow inputs for the season, (b) The riparian GW table remained in the soil zone throughout the season, and this location (T1E1) was a source of CH$_4$, (c) The toe slope position (T1E2) GW table dropped below the soil zone in late July, but recovered after a late season rain event. Early in the season, this landscape position produced CH$_4$ but gradually increased CH$_4$ uptake as the GW table declined, (d) The backslope (T1E3) GW table dropped below the soil zone in late June, and was a CH$_4$ sink the entire season, with maximum uptake at the end of July.



### 3.3. Environmental influences on measured CH₄ fluxes

Net $CH_4$ uptake was largest in dry soils, and a transition to net emission occurred around 38−43% VWC (Fig. 7). Upland $f_{CH4}$ was significantly correlated with VWC ($r^2 = 0.36$, $p < 0.001$), and the variability in magnitude of $CH_4$ uptake increased with decreasing VWC. Although soil $CH_4$ concentrations were not correlated with VWC, the influence of VWC on diffusivity was

5 associated with a significant relationship between upland $f_{CH4}$ and VWC. Maximum efflux occurred at 43% VWC and maximum uptake occurred at 4.7 % VWC. At low VWC substantial $f_{CH4}$ into the soil occurred. Efflux out of the soil occurred at high VWC (~40−50%), and near net zero $f_{CH4}$ was measured through the full range of VWC (1.4−64%; Fig. 7).

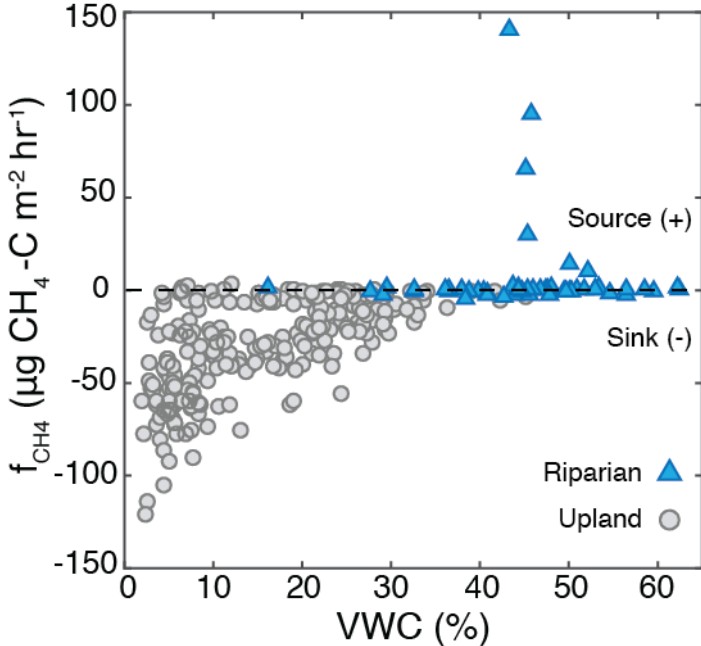

**Figure 7:** Measurements of $CH_4$ flux ($f_{CH4}$) and soil water content across all 32 sites for all sampling
dates. Magnitude of $f_{CH4}$ in the riparian area was not related to VWC, while magnitude and variability of $CH_4$ uptake in the uplands increased with decreasing VWC.

Methane fluxes were not significantly correlated with % $O_2$ at any depth, nor with soil temperature (Fig. A3). $CH_4$ uptake was
15 constrained to samples with 5 cm $O_2$ above 19 %, and generally increased with increasing 5 cm % $O_2$, with the largest between 20 and 21% $O_2$ (i.e., at ca. atmospheric levels). $CH_4$ efflux occurred even when 5 cm $O_2$ was 21% and up to 19.5 % $O_2$ at 20 cm. The largest $f_{CH4}$ (either into or out of the soil) occurred between 8° and 14 °C, and declined with higher or lower temperatures.



**3.4 Cumulative seasonal CH₄ fluxes and relationships to environmental variables and landscape position**

Cumulative seasonal CH$_4$ fluxes ($F_{CH4}$) ranged from -170 to -33 mg CH$_4$-C m$^{-2}$ in the uplands and from -0.98 to 3.12 mg CH$_4$-C m$^{-2}$ in the riparian sites, with one riparian location producing a relatively large $F_{CH4}$ of 232 mg CH$_4$-C m$^{-2}$ (Fig. 8). Rates of upland consumption generally increased through the season, and were consistent across sites until July when cumulative fluxes

began to diverge (Fig. 8). Although most environmental variables (bulk density, C:N, Cs$_{oil}$, Ns$_{oil}$, T$_{avg}$, ∂$^{13}$C, and ∂$^{15}$N ) were not significantly correlated with ln|F$_{CH4}$|$_{in}$, the average VWC of each site was negatively correlated with ln|FC$_{H4}$|$_{in}$ (r$^2$ = 0.32, p < 0.01).

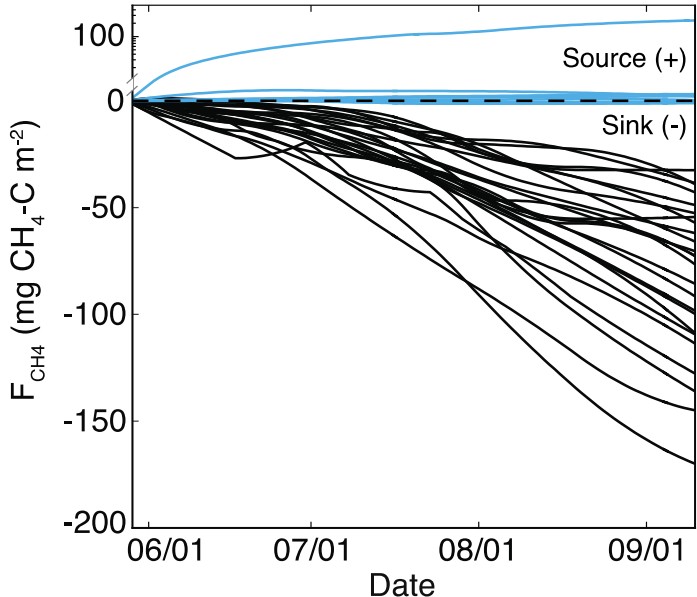

**Figure 8:** Cumulative CH$_4$ flux (F$_{CH4}$) for each site, riparian in blue and uplands in black. Most riparian
sites were near neutral, with one location being a source; all upland locations were CH$_4$ sinks.

We assessed the degree to which terrain metrics were correlated with environmental variables and F$_{CH4}$ in order to understand how the characteristics of a given landscape position could influence environmental variables and resulting total seasonal CH$_4$ fluxes (cumulative fluxes). Relationships between ln|F$_{CH4}$|$_{in}$ and terrain metrics were stronger with the 10 m DEM than the 3
15  m DEM, therefore relationships reported below and remaining analyses were conducted with the 10 m resolution DEM. Cumulative seasonal CH$_4$ influx (ln|F$_{CH4}$|$_{in}$) was regressed against average VWC (VWC$_{avg}$) and temperature of each site (T$_{avg}$), soil characteristics, and each terrain metric (Fig. A1 and A2). We used the strength of these individual relationships to determine which soils variables and terrain metrics to include in the multiple regression models. If variables were cross correlated, the variable with a stronger relationship with ln|F$_{CH4}$|$_{in}$ was retained, therefore individual regressions should not be
used as independent predictors. Five of the nine terrain metrics evaluated had significant relationships with ln|F$_{CH4}$|$_{in}$, including





elevation, EAC, DFC, topographic wetness index (TWI), and upslope accumulated area (UAA) (Fig. A1). Elevation, distance from creek (DFC), and elevation above creek (EAC) showed a positive relationship with $ln|F_{CH4}|_{in}$, meaning locations farther away or higher in elevation above the creek (e.g. toward ridges) had higher $ln|F_{CH4}|_{in}$ than near-creek sites. UAA and TWI had a negative relationship to $ln|F_{CH4}|_{in}$, and were positively related to $VWC_{avg}$ (UAA $r^2 = 0.27$, $p < 0.01$; TWI $r^2 = 0.43$, $p < 0.001$).

The negative influence of soil moisture on $ln|F_{CH4}|_{in}$ resulted in lower $ln|F_{CH4}|_{in}$ in locations with higher TWI or UAA. In summary, both landscape mediated relative water availability and local VWC explained net uptake of $CH_4$ across the watershed during the 2013 growing season.

### 3.5 Multiple regression model of cumulative upland CH₄ fluxes

Multiple regressions that included soils data explained up to 60% of the observed variability in $ln|F_{CH4}|_{in}$ (Table 3), showing

that although not as readily available, the addition of soils variables can improve modelling results. Although these models cannot be extrapolated to the watershed scale, including Nsoil improves the model performance by 12%, whereas including $T_{avg}$ and $\delta^{13}C$ only improves the model by 3 and 2% respectively (Fig. A3).

We created a spatially explicit model of upper Stringer Creek $ln|F_{CH4}|_{in}$ using the topographically based model with the best fit

(adjusted $r^2 = 0.47$, $p < 0.001$), and lowest BIC (Bayesian Information Criterion for model selection, where the model with the lowest BIC is preferred; Schwarz, 1978). This model included only TWI and elevation as parameters (Eq. 5, Table 3 & Fig. 9). The seasonal $CH_4$ uptake from the spatial model reached up to 2.1 kg $CH_4$-C ha$^{-1}$, averaged 0.77 kg $CH_4$-C ha$^{-1}$ and totaled 299 kg $CH_4$-C for the entire upland area (Table 3). We extrapolated the mean, median and maximum riparian $F_{CH4}$ to estimate contributions from riparian area (5 ha), resulting in a range of potential total riparian $F_{CH4}$ (Table 4). Even when using the

maximum riparian $CH_4$ efflux, the riparian emission offset < 4 % of total upland $CH_4$ influx, highlighting the strong role of upland uptake in the net landscape $CH_4$ balance.

$$ln|F_{CH_4}|_{in} = k_1 * TWI + k_2 * Elevation + k_3, \qquad (5)$$




**Figure 9:** Measured cumulative CH$_4$ influx (ln|F$_{CH4}$|$_{in}$) versus predicted ln|F$_{CH4}$|$_{in}$ for (a) the model that includes soils variables, and **c** the model that only used topographic variables. Grey lines are 1:1 lines for reference. The associated adjusted r$^2$ is shown with each model, details on model fit and coefficients are in Table 3. (b) Sampling locations and associated error for the soils model, (d) Map of ln|F$_{CH4}$|$_{in}$ across the upper Stringer Creek watershed showing results from the topographic model. Size of site symbols are scaled to their mean square error and color is associated with predicted flux shown in (a) and (b). (e) standard error from the topographic model.




## 4 Discussion

We utilized understanding of watershed hydrology processes at TCEF (Jencso et al., 2009; Jencso et al., 2010; Kelleher et al., 2017; Nippgen et al., 2015) to design a sampling campaign which captured $CH_4$ fluxes across environmental gradients that were characterized through topographic analysis, field observation, and hydrological measurements. This approach allowed

5 us to assess environmental influences on $CH_4$ fluxes; at the point scale, we examined the influence of environmental variables on observed $CH_4$ fluxes ($f_{CH4}$), at the intermediate scale, we identified functional landscape elements (riparian, upland, and the transition between them) which related to the direction and persistence of $f_{CH4}$, and at the landscape scale, we assessed the influence of topographic position on cumulative $CH_4$ fluxes ($\ln|F_{CH4}|_{in}$) in the uplands. Our observed average $f_{CH4}$ ($-28.5 \pm 25.1$ µg $CH_4$-C m$^{-2}$ h$^{-1}$, Table 2) was comparable to those of other temperate forests which range from -333 to 0.75 µg $CH_4$-C m$^{-2}$

h$^{-1}$ (mean: 32.9 µg $CH_4$-C m$^{-2}$ h$^{-1}$, standard error: 18; Dalal et al., 2008). We used observed relationships between $\ln|F_{CH4}|_{in}$ and topographic metrics to create multiple regression models of varying complexity to estimate the total watershed $F_{CH4}$. The average predicted $F_{CH4}$ from the spatially distributed model of upland $CH_4$ fluxes was similar to the extrapolated mean of measured $F_{CH4}$. This is partially due to our sampling approach which captured a range of landscape positions found at TCEF. It should be noted that simply extrapolating a mean flux from a measurement site or multiple measurement sites does not

capture the frequency distribution of similar landscape positions unless this is built into the sampling scheme (Vidon et al., 2015). Thus, we suggest capturing and/or modelling the spatial variability of landscapes is critical to estimating $CH_4$ consumption or efflux across landscapes.

### 4.1 How do environmental variables relate to $CH_4$ flux through the growing season and how does landscape structure relate to relative magnitude and direction of $CH_4$ fluxes across the landscape?

Research on soil-atmosphere $CH_4$ exchange has been conducted across a range of ecosystems (Smith et al., 2000; Castaldi and Fierro, 2005; Dalal and Allen, 2008; Luo et al., 2013), but assessing the spatial and temporal variability of $CH_4$ fluxes at the landscape scale has been limited. Studies focused on $CH_4$ oxidation have shown varied responses to commonly measured environmental variables such as soil moisture and temperature (e.g. Adamsen and King, 1993; Bradford et al., 2001; Price et al., 2004), nutrient variability (e.g., N species: Verchot et al., 2000; and dissolved organic carbon: Sullivan et al., 2013). In

addition to these physiological constraints, soil structure and texture create the physical landscape at the microbial scale by mediating how quickly soils drain and saturate, directly influencing transport and diffusion of substrates and $O_2$ (Dorr et al., 1993; Czepiel et al., 1995; Ball et al., 1997; von Fischer et al., 2009). Soil texture and nutrient status can be important in understanding the variability of $CH_4$ dynamics between ecosystems or dominant landscape units (Boeckx et al., 1997; Saari et al., 1998), but these factors did not have a significant influence on $CH_4$ uptake at the landscape scale. Although we did not find

relationships between soil characteristics and $CH_4$ uptake, small scale (cms–meters) variability in soil structure and organic matter can be particularly relevant in low moisture conditions, where even with similar values of VWC a range of soil moisture conditions (and therefore diffusivity) can occur.





Rates of both soil $CH_4$ production and consumption have been shown to increase with increasing temperature in laboratory studies (Bowden et al., 1998) and in field studies spanning wetlands, rice paddies (Bartlett and Harriss, 1993; Segers, 1998; Meixner and Eugster, 1999; Yvon-Durocher et al., 2014), spruce forests of Germany (Steinkamp et al., 2001), the Mongolian steppe region (Wu et al., 2010), and alpine grasslands (Wei et al., 2014). However, consensus has not been reached on the

5 relationship between $CH_4$ flux and temperature across ecosystems (Luo et al., 2013). In fact, several studies have shown limited temperature influence on daily and seasonal variability of $CH_4$ consumption in uplands soils (King and Adamsen, 1992; Del Grosso et al., 2000; Smith et al., 2000; Castaldi and Fierro, 2005; Shrestha et al., 2012; Imer et al., 2013). At TCEF, we did not find a simple relationship between $f_{CH4}$ and soil temperature. Early in the growing season, when soils were near saturation due to the recent snowmelt, both low soil temperatures and restricted gas phase transport were likely limiting $f_{CH4}$. As the

10 season progressed and temperatures increased, the largest range and magnitude of $f_{CH4}$ was observed, but these conditions coincided with increased diffusivity due to decreasing soil moisture, making the independent effects difficult to ascertain. These compounding seasonal factors in both riparian and upland settings, and the relatively low range of variability in soil temperature suggest our site is not an ideal location for assessing temperature effects on $CH_4$ fluxes. Given these caveats, our results do agree with findings from a study of temperature and moisture effects on methane consumption across ecosystem

types (Luo et al., 2013), which found maximum $CH_4$ uptake corresponded with average soil temperature (Fig. A3).

Depth to groundwater table, VWC and $O_2$ have been used to estimate soil redox conditions that are essential for methanogenesis (Fiedler and Sommer, 2000; Liptzin et al., 2011). As depth to water table increases, the volume in which oxidation can occur increases, thereby decreasing net $CH_4$ efflux (Moore and Roulet, 1993), yet similar to Fiedler and Sommer

(2000), we found that depth to groundwater table was not sufficient to predict the magnitude of riparian $CH_4$ efflux. High VWC is often associated with $O_2$ depletion (Silver et al., 1999), yet we measured near atmospheric $O_2$ even up to 60% VWC, similarly to Hall et al. (2012) who suggest that high VWC does not necessarily lead to depleted $O_2$, even when soil water content is above field capacity. Additionally, Teh et al. (2005) show that laboratory experiments varying $O_2$ does not result in significant changes in rates of methanogenesis. Based on these findings, we suggest that using VWC as a proxy for $O_2$

conditions should be used with caution when estimating biogeochemical fluxes reliant on redox conditions, and highlight the limited support for predictability of $CH_4$ efflux based on $O_2$.

Soil moisture has a strong influence on the microbial populations that drive methane cycling (Conrad, 1996; Potter et al., 1996; Smith et al., 2003; Luo et al., 2013; Du et al., 2015), but the differential response of methanotrophs and methanogens to soil

moisture status can make it difficult to find simple relationships between net $CH_4$ flux and VWC. The hydrologic landscape at TCEF is such that the groundwater dynamics are heavily influenced by topography (Jencso et al., 2009, 2010), which creates a range of soil moisture conditions across the uplands and a distinct riparian area that maintains a shallow water table through the growing season (Fig. 5 and 6). We assessed the direction, magnitude and seasonality of $f_{CH4}$ and determined the patterns



created by the soil moisture conditions influencing these fluxes functionally corresponded to riparian, transitional, and upland landscape elements.

Riparian zones are often characterized by high rates of biogeochemical cycling due to organic carbon availability, fluctuating
water tables and correspondingly variable redox conditions. At TCEF, soil in the riparian area is saturated during the snowmelt period, and the hydrologic connection to the uplands provides a downslope pulse of dissolved organic carbon (Pacific et al., 2010). This seasonal input of carbon could lead to increased methanogenesis, yet soil $CH_4$ concentrations remained relatively consistent throughout the growing season (data not shown). Despite this, and that the riparian locations sampled at TCEF maintained a water table throughout the season, these sites often exhibited little to no measurable $CH_4$ flux (Fig. 6, 7). These
low $CH_4$ fluxes are consistent with observations from other forest riparian areas, where much of the $CH_4$ produced deeper in the soil is oxidized before reaching the soil surface ((humid tropics: Teh et al., 2005; floodplain wetland: Batson et al., 2015; seasonally dry ecosystems: von Fischer and Hedin, 2002; Castaldi et al., 2006; riparian area: Vidon et al., 2015). The few large fluxes observed might be due limited sampling when gas wells were inundated (potentially missing ebullition events), fluxes of dissolved $CH_4$ through the groundwater to the stream channel (Itoh et al., 2007), recalcitrance of organic matter (Valentine
et al., 1994; Updegraff et al., 1995), or lack of sampling of fluxes from riparian vegetation, which can be an important transport process in wetlands (Whiting and Chanton, 1993; Shannon et al., 1996; Bridgham et al., 2013). Given these caveats, 115 samples over 13 weeks of sampling show that although riparian areas can be locations of high rates of biogeochemical cycling, large net emissions of $CH_4$ were not common among the riparian sites sampled at TCEF.

Transition zones, or boundaries between landscape elements can exhibit steep gradients in hydrologic conditions and nutrients (Hedin et al., 1998a). We determined that this was also true for $CH_4$ dynamics, which shifted from $CH_4$ efflux in the saturated soils of the riparian area to $CH_4$ uptake in the aerated soils of the uplands. Distinguishing the general boundary between riparian and upland landscape elements can be tractable using terrain metrics (here, the EAC threshold of 2 m), but accurately capturing the shifting spatial extent of the transition zone through time can be challenging (Creed and Sass, 2011). At TCEF, this required
direct measurement of the local groundwater table. The near net zero $F_{CH_4}$ in these transitional sites were a culmination of both $CH_4$ efflux and uptake rather than a consistent intermediate VWC that created near neutral fluxes throughout the season. We did observe near zero fluxes in the VWC range of 38−43% that are in accordance with VWC thresholds (32−44%) differentiating net $CH_4$ efflux from net uptake in other upland forests (Sitaula et al., 1995; Luo et al., 2013), but this intermediate VWC is likely a transient state that occurs in some parts of the landscape rather than being characteristic of a
landscape position throughout the season. We expect the transition zones could be particularly sensitive to climate variability, because the resulting changes in hydrologic dynamics could shift their boundaries and net $CH_4$ flux behaviour.

Flux of $CH_4$ into the soil ($f_{CH_4}$) was strongly mediated by local soil water content (Fig. 7 and A4), resulting in a seasonal pattern of $f_{CH_4}$ that was reflective of the snowmelt dynamics in this watershed. During and shortly after snowmelt, relatively high



upland VWC constrained $f_{CH4}$, and even resulted in a few small sources of $CH_4$ (Fig. 6). Low rates of $f_{CH4}$ could have been due to the combined effects of restricted diffusion of $CH_4$ (and $O_2$) into the soil, production of $CH_4$ deeper in the soil, and/or low temperatures. As the soil moisture state of the watershed decreased, gas-phase transport of $CH_4$ into the soil increased, microsites of potential methanogenesis decreased, and those combined effects increased $CH_4$ uptake through the growing

season (Fig. 6). Previous studies have suggested that there is an optimum water content range for $CH_4$ oxidation, below which methanotrophs become water stressed and consume less $CH_4$ (Adamsen and King, 1993; Torn and Harte, 1996; West and Schmidt, 1998; Dunfield, 2007). Here, we did not find a pronounced decrease in uptake at low water content; in fact, we observed our largest measured influx at an extremely dry site (Fig. 7), we note however, this was preceded by a rain event which might have influenced this $f_{CH4}$ measurement (Lohse et al., 2009). Net $CH_4$ consumption at low water content has been

documented in other systems, most notably in arid environments (savannas: Otter and Scholes (2000); desert soils: McLain and Martens (2005); shrublands: Castaldi and Fierro (2005), and in some temperate forests: Castro et al. (1995). At TCEF, the driest sites were not only the locations of the largest measured $CH_4$ uptake, but also showed the greatest variability in $f_{CH4}$, again highlighting the potential influence of small scale heterogeneity in soil texture and nutrient status.

**4.2 Prediction and scaling of CH₄ consumption using terrain analysis**

Greenhouse gases have been modelled using a range of frameworks including empirical (data-driven), mechanistic (process based), and atmospheric inverse modelling (see Blagodatsky and Smith, (2012), and Wang et al. (2012) for detailed reviews). Although these modelling efforts have significantly advanced our understanding of GHG dynamics at landscape to regional scales, most of them do not reflect spatial patterns (or variability) in the lateral redistribution of water (Tague and Band, 2001; Groffman, 2012). The spatial patterns of soil properties (Konda et al., 2010), microbial assemblages (Florinsky et al., 2004),

and resultant biogeochemistry influenced by landscape position and topography (Creed and Beall, 2009; Riveros-Iregui and McGlynn, 2009; Creed et al., 2013; Anderson et al., 2015) have been investigated and used to scale point observations to the larger landscape in a limited number of studies. Remote sensing and vegetation classification have also been suggested as empirical methods to scale $CH_4$ effluxes from wetlands to larger areas (Bartlett et al., 1992; Bubier et al., 1995; Sun et al., 2013), but similar remotely sensed scaling of soil $CH_4$ uptake is currently lacking.

We used an empirical model based on topographic indices to scale $CH_4$ fluxes from point measurements to the watershed scale. The extensive area of dry uplands consuming $CH_4$ (98% of watershed area), and low average production from the small riparian area resulted in a watershed net growing season sink up to $299 \pm 8$ kg $CH_4$-C (0.77 kg $CH_4$-C $^{ha-1}$). We found higher uncertainty in the near-stream area, this is likely due to the influence of higher TWI in locations that have an EAC above the riparian

threshold of 2 m (Fig. 9). These locations might behave more like the transitional areas which are saturated early in the season, and no longer have a groundwater table, or saturated conditions, later in the season. This spatially distributed model ($\ln|F_{CH4}|_{in}$ $\propto$ TWI and elevation) estimated a total net seasonal $CH_4$ uptake similar to the $CH_4$ uptake estimated by extrapolating the mean $F_{CH4}|_{in}$. This might partially be because the model did not capture the highest cumulative fluxes well, and had higher





standard error in the dry, high elevation locations (Fig. 9). The high frequency of landscape settings that experience drier conditions represent the disproportionate amount of the landscape which exhibits high net $CH_4$ uptake. Therefore, extrapolating a mean value to the entire watershed can bias estimates across watersheds. The use of central tendency and its effects on estimating GHG fluxes across landscapes was also highlighted with respect to $CH_4$ by Vidon et al. (2015), and has significant

implications for our understanding of the contribution of upland landscapes to regional and global $CH_4$ inventories.

Terrain analysis reflects the long-term conditions of a given location relative to its landscape setting. Lower VWC (at the point scale) or relative water availability (as represented by TWI at the landscape scale), corresponded to more $CH_4$ uptake, and are the most influential parameters at those respective scales due to their influence on microbial activity and soil diffusivity (Fig.

A4). Our modelling exercise demonstrates soil variables can aid in the explanation of $CH_4$ uptake (particularly at high $CH_4$ uptake), and suggests that we might be conservatively estimating $CH_4$ consumption given the lack of soil parameters in the spatially distributed estimate of $CH_4$ consumption (Equation 5).

Consistent with previous research on $CO_2$ fluxes at TCEF (Riveros-Iregui and McGlynn, 2009) and other studies (Duncan et

al., 2013; Vidon et al., 2014) our regression model results suggest that the topographic redistribution of water and the frequency distribution of relevant functional landscape elements should be considered in scaling exercises. These approaches may better reflect $CH_4$ dynamics in a variety of watersheds, such as locations where the riparian extent is proportionally larger and potentially offsets the upland $CH_4$ sink to a greater degree (Sakabe et al., 2016). Here, even if the maximum $F_{CH4}$ from the riparian area was used to estimate net efflux, it would have to comprise over 25% of the watershed area to offset the net $CH_4$

consumption in the uplands. As noted in a recent review by Bernhardt et al. (2017) it is critically important to perform these scaling exercises to determine the relative influences of point scale measurements on net watershed balances. These results highlight the importance of accounting for the upland $CH_4$ sink which can significantly offset high rates of methane production in riparian areas.

## 5 Conclusions

The strong gradients of water availability at TCEF impose both a direct (local) and indirect (distal / historic) effect on the microbial communities and physical transport processes regulating biogeochemical fluxes. We implemented a sampling design that utilized these hydrologic gradients to study the influence of landscape heterogeneity on watershed $CH_4$ fluxes. We determined that soil moisture was the dominant environmental influence on the direction of net $CH_4$ fluxes, and the magnitude of $CH_4$ uptake in the uplands due to its influence on soil diffusivity. Low nutrient status and limited range in soil temperature

could be responsible for the lack of a direct relationship between $C_{soil}$, $N_{soil}$ or $T_{avg}$ to $CH_4$ uptake, but likely contribute to the variability in observed $CH_4$ fluxes across the landscape.



Landscape elements can be useful in characterizing areas that behave similarly as net sources or sinks of $CH_4$, but the boundary between elements can shift as the landscape dries down or wets up. Although riparian areas can disproportionally contribute to net landscape biogeochemical fluxes, their area relative to the uplands made them a minor component of the $CH_4$ balance in upper Stringer Creek. Interestingly, there was limited support for a consistent seasonal trend in $CH_4$ effluxes in the riparian

area, while the uplands increased in sink strength as the growing season progressed.

The effect of soil moisture on $CH_4$ uptake led to an observable relationship between landscape structure and $CH_4$ flux. We used these relationships to create empirically derived multiple regression models with spatially distributed parameters. This allowed us to better visualize spatial patterns of fluxes and to extrapolate from measurement locations the watershed scale.

This is preferable to the use of central tendency which does not incorporate the frequency distribution of landscape settings relative to measurement locations. These findings contribute to the literature on the importance of spatial heterogeneity, and the lateral redistribution of water, and suggest that we could be significantly under predicting net watershed $CH_4$ sink strength if we do not account for spatial variability.

## 6. Data

Data can be found at https://www.hydroshare.org/resource/764a2568364a492183e548e7b0819551/



## 7. Appendix A

**Figure A1:** All topographic variables included in initial exploratory data analysis. If a set of topographic variables had a pearson correlation coefficient greater than 0.6, then the variable with a lower correlation with $\ln|F_{CH4}|_{in}$ was removed from the analysis. Significance is denoted by asterisks () < 1, (*) ≤ 0.05, (**) ≤ 0.01, (***) ≤ 0.001. Blue text indicates negative relationships and red indicates positive relationships. Histograms of each variable run diagonally and separate the correlation coefficients from the bivariate plots. Bivariate plots with filled symbols denote significant relationships. Aspect parameters were calculated in radians using Equation 1.



**Figure A2:** All environmental variables considered in the initial exploratory data analysis. If a set of variables had a pearson correlation coefficient greater than 0.6 the variable with a lower correlation with ln|F$_{CH4}$|$_{in}$ was removed. Significance is denoted by asterisks () ≤ 1, (*) ≤ 0.05, (**) ≤ 0.01, (***) ≤ 0.001. Blue text indicates negative relationships and red indicates positive relationships. Histograms of each variable run diagonally and separate the correlation coefficients from the bivariate plots. Bivariate plots with filled symbols denote significant relationships.





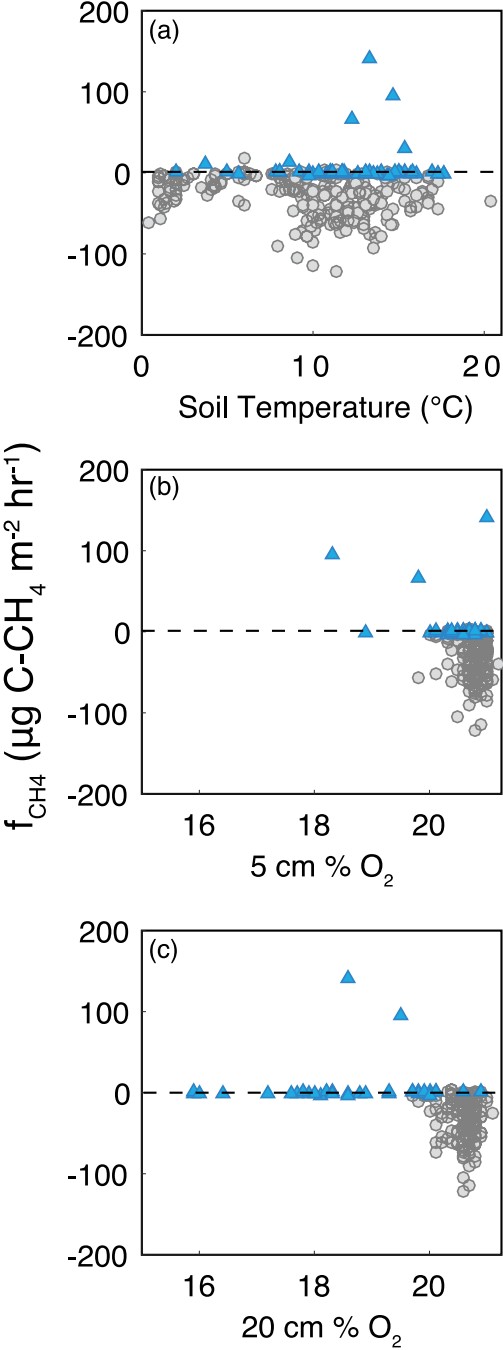

**Figure A3:** Bivariate plots of CH$_4$ flux (f$_{CH4}$) with (a) Soil temperature, (b) 5 cm %O$_2$, (c) 20cm % O$_2$. Circles are measurements from upland locations and triangles are from riparian locations.




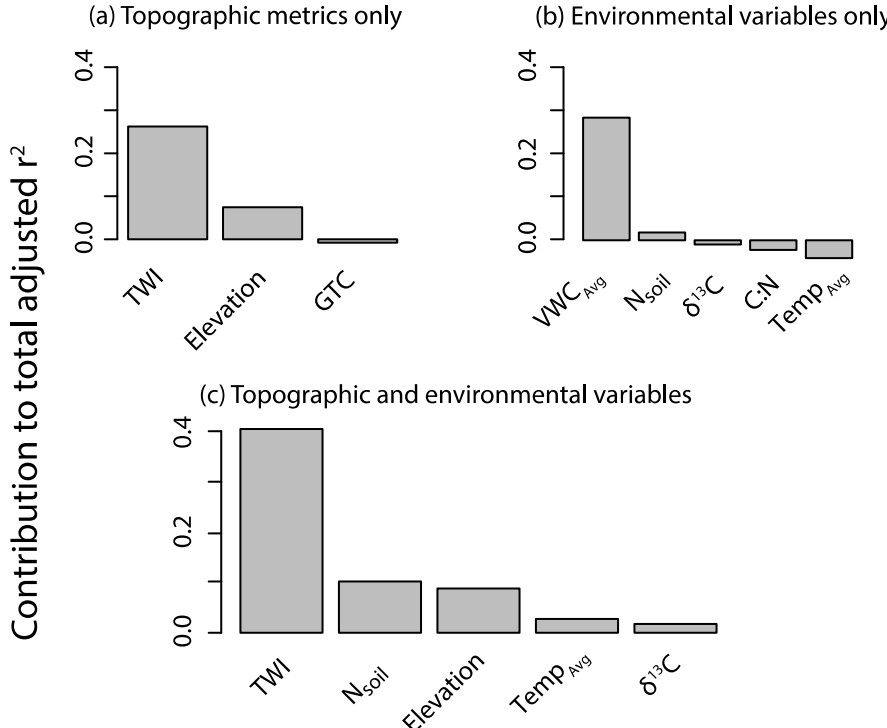

**Figure A4:** Results from the variable jack-knife analysis to determine which variables were the most important from a given parameter set: (a) topographic metrics (in final MLR), (b) soils variables only, and (c) topographic and soils variables (from grouped MLR). Variables included in each set are shown on the x-axis. The y-axis indicates the contribution to the total adjusted $r^2$ of the model when the variable is included in the model (the larger the number the more important the variable is to the model, and the higher its relative influence on $\ln|F_{CH4}|_{in}$. Given the penalty to the adjusted $r^2$ for additional variables, some variables did not contribute to model performance, and were not included in any of the models (e.g. $T_{avg}$, C:N).

## 8. Author Contributions

B.L.M. and J.E.D conceived of the initial project as project principal investigators. The project became a collaborative effort with K.E.K. who conducted most of the field work. J.E.D. led collection of the LiCor data for the diffusivity relationship and analyzed the gas samples. K.E.K. led data analysis and prepared the manuscript with contributions from her advisor B.L.M. and revisions from both co-authors.



## 9. Acknowledgments

This work was principally supported by NSF grant 1114392 awarded to J. E. Dore and B. L. McGlynn and an NSF GRFP fellowship awarded to K. E. Kaiser. Additional support to J. E. Dore from NSF EPSCoR Cooperative Agreement #EPS-1101342 is gratefully acknowledged. The authors appreciate logistical collaboration with the USDA Forest Service,

particularly H. Smith of the Rocky Mountain Research Station and C. Hatfield of the Lewis and Clark National Forest. We thank C. Allen, W. Avery, A. Birch, K. Brame, M. Burr, P. Clay, T. Covino, C. Dore, H. Dore, M. Dore, R. Edwards, J. Irvine, K. Jencso, R. Jones, L. Liang, T. Lorenzo, T. McDermott, A. Michaud, F. Nippgen, S. Ohlen, H. Wilson, E. Zignego, and M. Zimmer for field and/or laboratory assistance, and D. Urban for his suggestions on how to improve assessment of model performance. We would particularly like to thank E. Seybold for her critical contributions in the field and assistance with data

quality control.

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

**Table 1.** Average soil characteristics with 1 standard deviation in parentheses.

| Landscape Unit | Bulk Density ($g\ cm^{-3}$) | Total Porosity | C ($g\ cm^{-3}$) | N ($g\ cm^{-3}$) | Molar C:N |
|---|---|---|---|---|---|
| Uplands | 0.75 (0.17) | 0.65 (0.06) | 4.7 (1.9) | 0.17 (0.064) | 31.6 (7.8) |
| Riparian | 0.64 (0.47) | 0.76 (0.09) | 5.6 (1.3) | 0.41 (0.12) | 16.4 (2.2) |

**Table 2:** Methane flux statistics ($\mu g\ CH_4\text{-}C\ m^{-2}\ hr^{-1}$); SD = 1 standard deviation.

| Landscape Unit | Mean | SD | Median | Min | Max | Skewness |
|---|---|---|---|---|---|---|
| Uplands | -28.5 | 25.1 | -22.9 | -121 | 3.53 | -0.841 |
| Riparian | 6.54 | 24.9 | 0.186 | -4.44 | 141 | 4.17 |

**Table 3:** Coefficients of the parameters used to model cumulative seasonal influx ($\ln|F_{CH4}|_{in}$), and
statistical measures of model performance (all $p < 0.01$ ). MSE is the mean square error calculated from
15 the leave one out cross validation (Methods Section 4.8). TWI is unitless, and aspect is scaled from 0 to
1. Model types are differentiated by the inclusion of soils data to show how the lack of soils data in a
spatailly distributed estimate of upland $\ln|F_{CH4}|_{in}$ decreases goodness of model fit.

| | | Topography & Soil | Topographic Metrics Only |
|---|---|---|---|
| **Coefficients** | Intercept | -2.08 | 1.36 |
| | TWI | -0.236 | -0.210 |
| | Elevation (m) | $2.39 \times 10^{-3}$ | $2.01 \times 10^{-3}$ |
| | $N_{soil}$ ($g\ m^{-3}$) | -2.67 | - |
| | $\delta^{13}C$ (‰) | -0.162 | - |



|  |  |  |  |
|---|---|---|---|
|  | Average Temp (C°) | -0.096 | - |
| **Model** | Adjusted $r^2$ | 0.60 | 0.47 |
| **Performance** | BIC | 23.4 | 24.4 |
|  | MSE | 0.13 | 0.12 |

**Table 4:** Modeled and observed seasonal $CH_4$ uptake (separated by landscape element), as well as total, areally integrated seasonal $CH_4$ exchange. Observed mean and median of upland $F_{CH4}$ bracket the estimated average $F_{CH4}$ from the spatially distriuted topogrphic model. Total estimated riparian $CH_4$ effluxes are orders of magnitude smaller than uptake in the uplands.

|  |  | Cumulative Seasonal Flux (kg $CH_4$ -C $ha^{-1}$) | Total Seasonal Exchange (kg $CH_4$ -C) |
|---|---|---|---|
| **Modeled Upland Fluxes** |  | -0.77 | $-299 \pm 8.67$ |
| **Observed Upland Fluxes (area = 389 ha)** | Median | -0.73 | -282 |
|  | Mean | -0.83 | -322 |
| **Observed Riparian Fluxes (area = 5 ha)** | Max | 2.32 | 11.6 |
|  | Mean | 0.34 | 1.70 |
|  | Median | 0.02 | 0.11 |