# Peer review of "Landscape analysis of soil methane flux across complex terrain"

_Biogeosciences, 2017_

## Referee Comment (RC1) · Anonymous Referee #1 · 7 Feb 2018

The manuscript 'Landscape analysis of soil methane flux across complex terrain' reports soil-atmosphere methane exchange from a Rocky Mountain watershed. The manuscript focuses on understanding the environmental drivers of variations in methane exchange and the relation between these patterns and the topographic position of the sampling sites. Kaiser et al. found that upland sampling locations acted as net sinks for atmospheric methane and those close to the river could act as either weak sinks or sources. Whilst the controls on methane emission were unclear, an increase in methane uptake rates over the course of the growing season was attributed to the influence of decreasing soil water content on diffusion. Kaiser et al. proceed to show that this behaviour by the upland soils could be characterised by landscape scale descriptions of water availability. They conclude that such an approach is a desirable

avenue by which to up-scale from point measurements of methane uptake to upland soil budgets at the watershed scale.

The subject area addressed is of general interest and the manuscript is both well-written and presented. I have outlined a number of questions below that should be considered and addressed, followed by a few technical suggestions.

Methane fluxes were estimated using a gradient approach by measuring methane concentrations in gas wells and the effective diffusivity of methane (P8 L4 - L8). Effective diffusivity was characterised across the study site based on the relationship between flux measurements made with a static chamber and volumetric water content (Fig. 3). In doing so it is assumed that the total porosity and tortuosity of the soil is constant (P8 :17). How sensitive are the reported fluxes to this assumption given the actual variability in porosity reported in Table 1? Is the temperature sensitivity of the diffusion coefficient also accounted for here (Fig. A3)?

As soil physical properties were determined for each sampling location (P7 L1), did you test whether water-filled pore space (reflecting both soil porosity and water content) was a better proxy for diffusional constraints than volumetric water content (P20 L27 - 31)?

For how long or for how many cases did high water tables at the start of the growing season prevent measurement at the riparian sites (Fig. 6; P22 L13)? If flux measurements are lacking for periods when the water-table is within $\sim$ 5 cm of the surface, the general conclusion that methane emissions from these sites were rare (P22 L17) does not at first glance seem to be very robust given this apparent measurement bias.

Did this issue also influenced the ability to measure O2 concentration? The discussion of controls on methanogenesis (P21 L17 – L 26) suggests that water-table depth and O2 concentration are insufficient to explain the source behaviour of the riparian sites. However, I'm not sure that it is made sufficiently clear whether the authors are trying to generalise about the studied the system or are limiting themselves to instances when

there is a significant layer of soil exists above the water-table. This is important because we might expect the superficial soil layer, especially in mineral soils as this is where the majority of available carbon is found, to be the critical zone linking the balance between methanogenesis and methanotrophy to soil-atmosphere exchange (see Chamberlain et al., 2016 ,doi:10.1002/2015JG003283 for example).

I rather like the idea of using TWI as a spatially scalable proxy for the average hydrological conditions at a given point. It's clear from the text that the relationship between methane uptake and soil water content is well approximated by TWI (Fig. A1 & A2). As TWI is a function of the slope and up-slope drainage area at a given sampling location, the mechanistic link here is obvious (Section 4.2). Can you clarify what role you think elevation (above sea level?) plays in explaining methane uptake in best fit topographic model (Fig. 9)?

More generally, how applicable is the presented approach for other systems, for example, the humid tropical forests discussed in Section 4.1 which can vary between sink and source activity? The fact that topographic scaling isn't extended to the riparian zones seems to highlight a key limitation that should maybe be addressed more directly in the discussion and conclusion.

P9 Eq. 4: Should 'CH4' be 'DCH4' ?

P41 Table 1: Please check the units for C and N content.

P15 Fig. 6: Initially I found + and – for source and sink a bit distracting. Six panels separating flux and groundwater depth might look better?

P19 Fig. 9: Please check the panel labels a) through e) are correct and consistent in the legend. Also remove the extra bracket following units in on the x-axis.

---

## Referee Comment (RC2) · Anonymous Referee #2 · 11 Feb 2018

The manuscript presents an interesting work about landscape analysis of soil CH4 flux across a range of landscape positions (riparian and upland). This study aims to identify how topographic metrics-mediated environmental variables influenced watershed scale CH4 fluxes during the growing season. The authors found that riparian sites had near zero CH4 flux, while upland had significant CH4 uptake, which significantly correlated with topographic metrics. This study demonstrates the importance of spatial heterogeneity and the lateral redistribution of water on watershed net CH4 flux. It also points out the need of estimating CH4 fluxes across complex terrain through modeling the spatial variability of landscapes.

The objectives are clear and the methodological approaches are sufficient to answer the questions posed in the introduction and to justify the conclusions. The manuscript

is clear and easy to follow, with clear results and streamlined interpretation. The topic is well in line with the scope of the journal and the overall quality is also good.

Please find below specific comments:

Page 1, Lines 18-20 It is better to point out the time period that this finding was observed. Since the temperature and other parameters may change with time and thus the associated uptake/emission point (38

Page 2, Lines 10-12 Restructure the sentence to make it clear.

Page 2, Line 15 Better to rewrite this sentence.

Page 3, Line 25 Here, it is better to see the range of the gentle to steep gradient slopes.

Page 6, Lines 10-12 In the Eq.1, you described "slope=$\cos\theta$", and then "...where ...$\theta$ is slope" "$\theta$ is local slop". I am confusing about the definition of $\theta$ whether $\theta$ or $\cos\theta$ is slope? I would like to know if $\theta$ means the angle of the surface to the horizontal. If it is, doesn't slope equal to $\tan\theta$?

Page 7, Line 4 I do not understand why n=32, since you have 32 sampling sites and three soil sampling layers.

Page 9 How sensitive is the method for fCH4 calculation? There is a big variation in effective soil diffusivity as the VWC lower than 10

Page 10, Line 8 "CH4 fluxes" instead of "fCH4 fluxes".

Page 10, Line 9 "fCH4 and environmental variables" instead of "fCH4 measurements and environmental variables"

Page 12, Line 15 Change "Fig.4" to "Fig.4a"

Page 14, Lines 2-4 Change "Fig.4" to "Fig.4b"

Page 18 Lines 1-2 Place the definition of DFC to where it occurs the first time.

Page 18, Line 12 Should be Fig. A4

Page 18, Line 18 Should be Table 4

Page 19, Figure 9 In the legend, please check the panel letters to make sure they indicate the right figures.

---

## Author Comment (AC1) · 15 Mar 2018

The manuscript 'Landscape analysis of soil methane flux across complex terrain' reports soil-atmosphere methane exchange from a Rocky Mountain watershed. The manuscript focuses on understanding the environmental drivers of variations in methane exchange and the relation between these patterns and the topographic position of the sampling sites. Kaiser et al. found that upland sampling locations acted as net sinks for atmospheric methane and those close to the river could act as either weak sinks or sources. Whilst the controls on methane emission were unclear, an increase in methane uptake rates over the course of the growing season was attributed to the influence of decreasing soil water content on diffusion. Kaiser et al. proceed to show that this behaviour by the upland soils could be characterised by landscape scale descriptions of water availability. They conclude that such an approach is a desirable avenue by which to up-scale from point measurements of methane uptake to upland soil budgets at the watershed scale.

The subject area addressed is of general interest and the manuscript is both well written and presented. I have outlined a number of questions below that should be considered and addressed, followed by a few technical suggestions.

AC: We thank Referee #1 for positive and helpful comments. We endeavor to answer each of the Referee's queries in line below.

Methane fluxes were estimated using a gradient approach by measuring methane concentrations in gas wells and the effective diffusivity of methane (P8 L4 - L8). Effective diffusivity was characterised across the study site based on the relationship between flux measurements made with a static chamber and volumetric water content (Fig. 3). In doing so it is assumed that the total porosity and tortuosity of the soil is constant (P8 :17). How sensitive are the reported fluxes to this assumption given the actual variability in porosity reported in Table 1? Is the temperature sensitivity of the diffusion coefficient also accounted for here (Fig. A3)?

AC: With regard to the first question about the sensitivity of the flux measurements to porosity – it is not possible to analytically determine the impact of variable porosity on our derived fluxes, because the model derivation does not include a porosity term. We do not actually assume that porosity is constant; rather, we recognize that landscape variability in porosity is a contributor to the overall variability around our empirical exponential model fit of diffusivity to VWC. The 95% confidence intervals on the model fit are shown in Fig. 3, and these indicate the level of variability in diffusivity that can be expected from variability in porosity (and tortuosity), combined with measurement error. This variability in effective diffusivity leads to proportional variability in flux estimates, but the absolute values of errors in fluxes are also dependent on the magnitudes of the $CH_4$ gradients. We do mention that our model is mathematically equivalent to an exponential fit of diffusivity to air-filled porosity at constant total porosity (P9 L16-18).

As to the second question, the temperature sensitivity of the soil methane effective diffusivity $D_s$ is explicitly considered in our model; as shown in Fig. 3, we model ($D_s/D_o$) as a function of VWC, where $D_o$ is the free-air diffusivity of $CH_4$ at the measured temperature and barometric pressure. For each site occupation, temperature and pressure are used to calculate $D_o$ according to Massman (1998), and then $D_s$ is calculated from the model fit. We had neglected to mention this step in our Methods but have now done so (P9 L12-13) and have added the Massman (1998) reference (P36 L1-2).

As soil physical properties were determined for each sampling location (P7 L1), did you test whether water-filled pore space (reflecting both soil porosity and water content) was a better proxy for diffusional constraints than volumetric water content (P20 L27 - 31)?

AC: Thank you for the suggestions, we did not test this, because porosity was only measured at a subset of sites (n = 18, P7 L10). We have removed "soil texture" from P20 L27 to prevent confusion over whether porosity and soil texture were measured across all sites. We should also point out that one cannot determine the porosity of the soil directly above a gas well without disturbing the soil. Hence, collection of soil cores can only be done within the general vicinity (within a couple of meters) of the wells (P7 L2). Where replicate cores were pulled, the differences in their porosities ($\Delta\phi$ = 0.02 to 0.14; data not shown) were on par with the variability in porosity across the sites (see Table 1). Thus, we don't believe that using a diffusivity model based on air- or water-filled pore space (which would require knowledge of porosity over every gas well) would produce results with any less error than that indicated by the 95% confidence intervals in Fig. 3. In addition, we did try several models based on air-filled porosity over a range of fixed porosity values (P9 L18-20), and none fit the data better than the simple exponential fit of diffusivity to VWC.

For how long or for how many cases did high water tables at the start of the growing season prevent measurement at the riparian sites (Fig. 6; P22 L13)? If flux measurements are lacking for periods when the water-table is within ~5 cm of the surface, the general conclusion that methane emissions from these sites were rare (P22 L17) does not at first glance seem to be very robust given this apparent measurement bias.

AC: There were 26 missing measurements in the riparian area due to presence of water in the gas wells early in the growing season. When 5 cm gas wells were flooded, we frequently observed water ponding on the ground, indicating extremely low diffusivity. Ebullition under these conditions may be possible, but there is only a very short window of time following snow melt during which the water table lies between the surface and 5 cm depth, limiting the time for $CH_4$ to accumulate (see Fig. 6b). Furthermore, we did conduct LiCor flux chamber deployments (data not shown) at three riparian locations when wells were flooded following snow melt. Though we could not determine diffusivities because we could not measure gradients, the measured $CH_4$ effluxes were 3.58, 4.37 and 39.49 µg $CH_4$-C m$^{-2}$ hr$^{-1}$. These values are well within the range observed over the entire study (see Fig. 7). Hence, it does not appear likely that we missed large effluxes during the early season, and if there were missed bursts of $CH_4$ emission they would not likely have been of sufficient duration to skew the cumulative seasonal results. Additionally, relatively low available organic carbon, low temperature and/or extremely low diffusivity would all contribute to limitation of $CH_4$ efflux during the snowmelt period.

Did this issue also influenced the ability to measure O2 concentration? The discussion of controls on methanogenesis (P21 L17 – L 26) suggests that water-table depth and O2 concentration are insufficient to explain the source behaviour of the riparian sites. However, I'm not sure that it is made sufficiently clear whether the authors are trying to generalise about the studied the system or are limiting themselves to instances when there is a significant layer of soil exists above the water-table. This is important because we might expect the superficial soil layer, especially in mineral soils as this is where the majority of available carbon is found, to be the critical zone linking the balance between methanogenesis and methanotrophy to soil-atmosphere exchange (see Chamberlain et al., 2016 ,doi:10.1002/2015JG003283 for example).

AC: Yes, flooding of the wells precluded reliable measurements of all gases. Hence, it is possible that $O_2$ levels at flooded riparian sites declined during the snowmelt, but the flooded period is brief in this system (intermittent flooding over the course of a few weeks). We highlight that the $O_2$ concentration and water table height were not sufficient to predict the *magnitude* of $CH_4$ efflux (Fig. A3c), and edited the paragraph (P21 L26) to clarify.

We agree that the shallow soil (and it's dynamic $O_2$ conditions, and carbon availability) mediates the production and consumption of $CH_4$, but we hesitate to overgeneralize here. The environment studied by Chamberlain et al. was flooded pastureland, with relatively high organic carbon in the near-surface soils and relatively poor drainage (hence measurements of dissolved oxygen in the groundwater, rather than percent $O_2$ in unsaturated soil as in our study). Our semiarid subalpine forest location is relatively low in soil organic carbon and for the most part quite well drained (added this note P21 L21). Only within the riparian corridor, and only for a brief time each season, do conditions favor low $O_2$ and net efflux of $CH_4$.

Moreover, we have shown that emissions from the small riparian area do little to offset the $CH_4$ uptake of the much more extensive upland soils, making the ecosystem balance between methane consumption and production reflective of the proportion of the landscape that is riparian versus upland. We suspect that, even within subalpine forests, as one moves to watersheds where riparian areas comprise a larger fraction of the landscape, the role of $CH_4$ emissions will become more important in offsetting the upland soil $CH_4$ sink. However, before predictive capability can be realized in this regard, we recognize that we must understand the spatial heterogeneity of $CH_4$ exchange within these riparian zones close to as well as we have been able to do for the upland portions of the landscape. That is a big challenge because, as we have previously seen with the response of ecosystem $CO_2$ fluxes to temperature and precipitation, predictive relationships that arise within complex terrain may be absent in flat terrain (see W.M. Reyes et al. 2017, Glob. Biogeochem. Cycles 31:1306).

I rather like the idea of using TWI as a spatially scalable proxy for the average hydrological conditions at a given point. It's clear from the text that the relationship between methane uptake and soil water content is well approximated by TWI (Fig. A1 & A2). As TWI is a function of the slope and up-slope drainage area at a given sampling location, the mechanistic link here is obvious (Section 4.2). Can you clarify what role you think elevation (above sea level?) plays in explaining methane uptake in best fit topographic model (Fig. 9)?

AC: Elevation might be reflecting a combination of factors such as its influence on temperature, or the differences in mineral type with elevation which lead to differences in soil chemistry and pH. This variability could lead to differences in the soil microbial communities, in particular the microorganisms responsible for atmospheric methane oxidation.  It is noteworthy that microbial communities have been shown to significantly differ between higher and lower upland soils within this watershed (see Du et al. 2015). Thank you for pointing this out, we've added a few sentences to the discussion (P24 L18-22).

More generally, how applicable is the presented approach for other systems, for example, the humid tropical forests discussed in Section 4.1 which can vary between sink and source activity? The fact that topographic scaling isn't extended to the riparian zones seems to highlight a key limitation that should maybe be addressed more directly in the discussion and conclusion.

AC: Thank you for pointing this out, we have incorporated limitations of this method in the discussion (P23 L31) and in the conclusions (P25 L20). Again, we hesitate to overgeneralize (see above), but the approach could be applied in other locations. If we were to test this in energy limited locations the terrain metrics that are of most importance could change. The complexity of the terrain might be of more importance than the water limitation of the system because even in a humid environment, water redistribution would still occur, leading to spatial heterogeneity in soil moisture.

P9 Eq. 4: Should 'CH4' be 'DCH4' ? AC:  Yes, Replaced 'CH4' with 'DCH4'

P41 Table 1: Please check the units for C and N content. AC: We corrected units in the table and in the text (P7 L7) to be in (mg cm-3).

P15 Fig. 6: Initially I found + and – for source and sink a bit distracting. Six panels
separating flux and groundwater depth might look better?
AC: With respect to the Referee, we have tried different representations of these data, and believe that being able to directly compare the data in one figure is clearer than additional panels.

P19 Fig. 9: Please check the panel labels a) through e) are correct and consistent in
the legend. Also remove the extra bracket following units in on the x-axis. AC: Edited the panel labels in the text and the extra bracket.

**Anonymous Referee #2**

The manuscript presents an interesting work about landscape analysis of soil CH4 flux across a range of landscape positions (riparian and upland). This study aims to identify how topographic metrics-mediated environmental variables influenced watershed scale CH4 fluxes during the growing season. The authors found that riparian sites had near zero CH4 flux, while upland had significant CH4 uptake, which significantly correlated with topographic metrics. This study demonstrates the importance of spatial heterogeneity and the lateral redistribution of water on watershed net CH4 flux. It also points out the need of estimating CH4 fluxes across complex terrain through modeling the spatial variability of landscapes.
The objectives are clear and the methodological approaches are sufficient to answer the questions posed in the introduction and to justify the conclusions. The manuscript is clear and easy to follow, with clear results and streamlined interpretation. The topic is well in line with the scope of the journal and the overall quality is also good.
AC: We thank Referee #2 for reviewing our manuscript favorably and for providing helpful specific comments.  We address these individually below.

Please find below specific comments:
Page 1, Lines 18-20 It is better to point out the time period that this finding was observed.
Since the temperature and other parameters may change with time and thus the associated uptake/emission point (38
AC: The lines in question refer to median values across the entire study, so we did not mention the time period here, but we have added this time frame at P1 L15.  In our Methods we do define the "season" for cumulative data as May 29th to September 12th (P9 L21).

Page 2, Lines 10-12 Restructure the sentence to make it clear. AC: Edited sentence to be clearer.

Page 2, Line 15 Better to rewrite this sentence. AC:  Restructured sentence.
Page 3, Line 25 Here, it is better to see the range of the gentle to steep gradient slopes. AC: In the upper Stringer Creek watershed slopes range from 0-66%, but the very steep slopes are quite rare, and we believe the mean slope is a better descriptor of the watershed as a whole. Moreover, the topographic map of the watershed (Fig. 1) gives the reader an idea of the overall variability of slope across the landscape.

Page 6, Lines 10-12 In the Eq.1, you described "slope=cosθ", and then "…where…θ is slope" "θ is local slope". I am confusing about the definition of θ whether θ or cosθ is slope? I would like to know if θ means the angle of the surface to the horizontal. If it is, doesn't slope equal to tanθ?
AC: Θ is the percent slope, the individual effect of slope is determined using the percent slope raster, but the combined effects of aspect and position-on-slope were determined using the vector of covariates. We had separated them to make the individual components more clear,  but have re-written the equation as one vector of co-variates as written in Clark 1990 (P6 L10).

Page 7, Line 4 I do not understand why n=32, since you have 32 sampling sites and three soil sampling layers. AC: Added sites to the descriptor to designate that the 32 refers to the 32 sites.

Page 9 How sensitive is the method for fCH4 calculation? There is a big variation in effective soil diffusivity as the VWC lower than 10
AC: We refer the Referee to our replies to Referee #1 above on this point.  The combination of porosity variability across the landscape and measurement error lead to uncertainty in diffusivity as delineated by 95% confidence intervals in Fig. 3.  Fluxes will be uncertain proportionally with this uncertainty in diffusivity, and with the absolute value of the measured $CH_4$ gradient.

Page 10, Line 8 "CH4 fluxes" instead of "fCH4 fluxes". AC: Edited

Page 10, Line 9 "fCH4 and environmental variables" instead of "fCH4 measurements and environmental variables" AC: Edited

Page 12, Line 15 Change "Fig.4" to "Fig.4a" AC: Edited

Page 14, Lines 2-4 Change "Fig.4" to "Fig.4b" AC: Edited

Page 18 Lines 1-2 Place the definition of DFC to where it occurs the first time. AC: Edited

Page 18, Line 12 Should be Fig. A4 AC: Edited

Page 18, Line 18 Should be Table 4 AC: Edited

Page 19, Figure 9 In the legend, please check the panel letters to make sure they indicate the right figures. AC: Adjusted labels in the figure caption, thank you.